# Calculation Model of Mechanical and Sealing Properties of NiTi Alloy Corrugated Gaskets under Shape Memory Effect and Hyperelastic Coupling: I Mechanical Properties

**DOI:** 10.3390/ma15144836

**Published:** 2022-07-12

**Authors:** Lingxue Zhu, Yang Liu, Mingxuan Li, Xiaofeng Lu, Xiaolei Zhu

**Affiliations:** 1Department of Mathematics, Jinling Institute of Technology, Nanjing 211169, China; zlx1987@jit.edu.cn; 2School of Mechanical and Power Engineering, Nanjing Tech University, Nanjing 211816, China; liuyang-pro@njtech.edu.cn (Y.L.); lmx9989w@njtech.edu.cn (M.L.)

**Keywords:** NiTi alloy corrugated gasket, shape memory effect, mechanical property, contact stress prediction

## Abstract

NiTi alloy’s shape memory effect provides additional restoring force under temperature loads, making it an ideal material for gaskets. However, its yield stress is too large to form the initial seal. In this paper, by combining the advantages of corrugated structure and NiTi alloy’s shape memory effect, a NiTi alloy corrugated gasket is proposed. Its mechanical properties were studied using experiments and the finite element method. The influences of geometric parameters on gasket performance were discussed. The results show that the shape memory effect can greatly improve the contact stress of gaskets. The corrugation can effectively reduce the pre-tightening force. The contact stress of NiTi alloy corrugated gasket is significantly affected by plate thickness, gasket height, and corrugation pitch and shows a high nonlinear characteristic. The proposed finite element method (FEM) and the gasket contact stress prediction model are accurate and engineering available.

## 1. Introduction

As a removable connection structure in process equipment, bolted flange connection is widely used in the chemical industry, nuclear energy, medicine, and other fields [1]. The failure of the connection system leads to leakage, which produces huge economic losses and causes serious casualties. As one of the three major causes of accidents in the chemical industry, leakage has been widely concerned [2]. As the core element of sealing in bolted flange connection, the gasket’s performance seriously affects the safety of the equipment. In recent years, process equipment was developed towards high parameterisation, multi-function, and large-scale [3]. This also puts forward higher requirements for sealing performance of equipment connection. The service environment is becoming harsher, and the sealing elements are required to have good environmental adaptability. High parametrisation requires that the sealing elements have high compression resilience. At the same time, the sealing gasket needs to have high reliability and can deal with the leakage of the connecting joint caused by temperature change or load fluctuation. Therefore, the study of new sealing gasket materials or new sealing structures can not meet the requirements above only from the perspective of new sealing materials or sealing structures.

Metal corrugated gaskets are widely used in the chemical industry, nuclear energy, and some other fields because of their good compression resilience and high line sealing specific pressure. Shape memory alloy has excellent mechanical properties. Scholars studied the effect of the treatment on mechanical properties [4,5,6,7,8]. Shape memory alloy is a sealing material with great potential. Its pseudoelasticity can avoid the plastic deformation of the gasket under short-term overload action and effectively ensure the spring back amount of the gasket. The shape memory effect can effectively guarantee the contact stress of the sealing surface of the gasket under operating conditions. Therefore the shape memory alloy gasket can effectively improve the reliability of bolt flange connection sealing under fluctuating load conditions.

However, the high hardness of the shape memory alloy makes it difficult for the shape memory alloy flat gasket to exert its advantages. Therefore, combining the structural advantages of corrugated gaskets with the material advantages of shape memory alloys is the key to solving the problem of high-parameter equipment to achieve high-reliability sealing.

In order to meet the increasingly stringent sealing requirements, scholars studied several kinds of memory alloy gaskets according to different sealing conditions. Tanaka et al. [9] first proposed to apply shape memory alloy to flange sealing connection. They proposed a new type of sealing and winding gasket with shape memory alloy as the skeleton. Its V-shaped winding belt adopted the shape memory alloy material, and the filler was graphite or some other soft materials. The shape memory effect and pseudoelasticity of the V-belt were used to compensate for the loss of sealing compression load caused by external factors. Efremov [10] introduced shape memory alloy into flange sealing connection at ASME annual meeting in 2006 and put forward relevant patents [11,12,13] in the following three years. Through different elements combination and treatment, the phase transition temperature and memory effect of shape memory alloy could be changed to meet the requirements of flange connection under different working conditions and improve the sealing performance. Reference [14] showed a flange sealing connection with a shape memory alloy ring gasket. Only the bolts needed to be tightened initially and then heated to above room temperature during assembly. Due to the memory effect, the sealing ring would return to the original memory shape. Therefore, it was not necessary to provide any high preload with bolts. Tan [15] et al. designed a self-tightening NiTi alloy double corrugated composite gasket. NiTi corrugated gasket and H-shaped metal plate were pressed through flange and contacted with flexible graphite when the gasket works. Flexible graphite provided the main compression and improved the sealing performance. Li [16,17] et al. developed a type of NiTi shape memory alloy disc gasket, which can reduce the possibility of bolt loosening caused by wind or electromagnetic effects in the electric transmission line.

Experiments and numerical calculations were used to further study the sealing performance and mechanical properties of shape memory alloy gasket. Tatsuoka et al. [18,19] studied NiTi shaped memory alloy gasket and compared it with aluminum and 304 flat gaskets. The NiTi alloy flat gasket had a smaller elastic modulus and higher yield strength. The NiTi alloy flat gasket had a better sealing performance, especially when installed at low temperature when the gasket stayed in the martensitic state. In the temperature cycle test, the sealing effect of the NiTi alloy gasket declined. However, the sealing compression force was still good. Y Takagi et al. [20] calculated the effect of temperature change on the sealing performance of NiTi shape memory alloy gasket pipe flange by using the coefficient of thermal expansion. Considering the temperature dependence of mechanical properties of NiTi alloy, the temperature effect on sealing was evaluated. Zhan et al. [21] explored the preparation of NiTi shape memory alloy corrugated pad and studied the effect of aging treatment on the mechanical properties of NiTi corrugated pad. The experimental results showed that the gasket had the best mechanical properties when it was aged at 500 °C for 60 min. The research results of Lu et al. [22,23] further prove that under static sealing conditions, appropriate aging heat treatment methods and reasonable operating temperature may be why shape memory alloy flat gasket has better compression resilience than the ordinary metal gasket. In addition, Nazim et al. [24,25] used the advantages of shape memory and hyperelastic behavior of shape memory alloy studs to compensate for the load loss caused by creep and heat exposure of gaskets in bolted components. In addition, Niccoli [26,27,28] studied the sealing performance of shape memory alloy rings in accelerators by finite element simulations and experimental measurements and found that the use of shape memory alloy could result in noticeable advantages. Lagoudas [29] analysed the contact pressure properties of shape memory alloy pipe couplers based on experiments and finite elements, and the NiTi coupler demonstrates a high contact pressure.

The deformation process of the shape memory alloy gasket in the bolt flange structure and its phase transition mechanism is shown in Figure 1. The first stage was the gasket installation stage. The gasket was installed between the two flanges and compressed by the flanges under the bolt load. During the installation process, the shape memory alloy gasket was in a martensitic state. Under the bolt load, martensite detwinning occurred, that is, the transition from multiple martensite variants to single martensite. The second stage was the operation stage. The gasket deformed under the internal pressure and the temperature load. Firstly, under internal pressure, the two flange surfaces separated, and the gasket resiled. Secondly, the material of the flange, bolt, and gasket softens at the operating temperature. At the same time, the gasket resiled due to expansion. Therefore, the deformation of bolted flange joint was more complex in the operation stage. For the NiTi alloy gasket, because of its shape memory effect and pseudoelasticity, the deformation process of the gasket was accompanied by a complex phase transition. The transition from single martensite to multiple martensites occurred in some parts of the shape memory alloy under internal pressure. At the operating temperature, both martensites transformed to austenite. Additionally, due to the restraint effect of the upper and lower flanges, some austenite underwent a stress-induced martensitic transition. Therefore, the phase transition process of the shape memory alloy gasket was more complex under operating conditions. There was interaction among the multiple phase transition processes. These complex phase transitions seriously affected the mechanical properties of gaskets.

In summary, this paper focused on NiTi alloy corrugated gasket. The deformation mechanism of NiTi corrugated gasket in its service was explored. The influence of structural parameters on the gasket’s mechanical properties was revealed. The calculation model of the compression-recovery performance of NiTi corrugated gasket was established. It provided the theoretical and data support for the further application of NiTi alloy corrugated gasket.

## 2. Materials and Methods

### 2.1. Experiment

The manufacturing process of the NiTi alloy corrugation sample is shown in Figure 2. The first step was to cut a NiTi alloy strip with a thickness of 0.3 mm. The second step was to press the strip into a corrugated sample with a mould. The third step was to put the mould containing NiTi alloy corrugated strip into an atmosphere-protected heat treatment furnace and set the heating temperature to 500 °C [21]. The holding time was 60 min. The heating rate was 4 °C/min. Then, the sample was cooled down to room temperature in the furnace after heat preservation and removed from the mould. Finally, the sample was put into the heat treatment furnace under atmosphere protection again, rising to 150 °C for 10 min, then cooled to room temperature in the furnace [21]. The sample was prepared, as shown in Figure 2f. The mould was made of H13 steel. The bolt was S304 stainless steel. The mould and the NiTi alloy corrugation sample are shown in Figure 3a,b, respectively. The composition of the NiTi alloy sheet is shown in Table 1. The phase transition temperature of the NiTi alloy sheet was measured by Differential Scanning Calorimetry (DSC) method. The starting temperature of martensite transition (Ms), the ending temperature of martensite transition (Mf), the starting temperature of austenite transition (As), and the ending temperature of austenite transition (Af) are 19.96 °C, 53.03 °C, 62.93 °C, and 77.23 °C, respectively.

The NiTi alloy gasket needed to undergo a complex phase transition process in service. The experimental procedure was drawn up according to the service process of the NiTi alloy gasket.

In the first step, a displacement load was applied to the corrugated sample at room temperature, and the maximum compression was 30% of the sample height to test the mechanical properties of NiTi alloy during the detwinning process. In the second step, when the maximum displacement was reached, the load was retained for 5 min. Then the unloading was performed to test the reversibility of the NiTi alloy detwinning process. In the third step, holding the position of the upper and lower loading head, start to heat the sample to 110 °C, which was greater than the end temperature of the NiTi alloy austenitic transition. The heating rate was 3 °C/min, and the temperature was held for 10 min to test the NiTi alloy austenitic transition and stress-induced martensitic transition process. The experimental instrument is shown in Figure 4.

### 2.2. Finite Element Analysis Method of NiTi Alloy Corrugation

ABAQUS/Standard was used to simulate the NiTi alloy corrugated specimens’ mechanical performance test process, study the phase transition characteristics during the loading process, and reveal the corrugated specimens’ deformation mechanism.

The constitutive model of NiTi alloy and UMAT subroutine is referred to in the literature [21]. The mechanical properties of the NiTi alloy sheet are shown in Table 2. The geometric model of the NiTi corrugation is shown in Figure 5a. C3D8R element was used in the corrugated sample mesh, and hourglass reinforcement control was selected to ensure the element stiffness. The element number in the thickness direction of the corrugated plate was 3, and the number of the whole elements was 272,152. The mesh model is shown in Figure 5b. The upper and lower loading heads were simplified as rigid bodies. The rigid plane and the NiTi alloy corrugated specimen were contacted by penalty function, with a friction coefficient of 0.15. The lower loading head was fixed, the upper loading head was controlled by displacement, and its boundary condition was U1 = U3 = UR1 = UR2 = UR3 = 0. The initial temperature field was set at 20 °C.

There were three analysis steps in the calculation. The first step was the compression process. The upper loading head moved downward in the U2 direction, and the maximum displacement was 30% of the height of the corrugated sample. The second step was the recovery process. The upper loading head was unloaded upward along the U2 direction to 0 N. The third step was the heat recovery process. The upper and lower loading heads were fixed, and the corrugated sample’s temperature field was set at 110 °C in the predefined field.

### 2.3. Finite Element Analysis Method of NiTi Alloy Gasket

According to the deformation mechanism of the NiTi corrugated specimen, the shape memory effect and pseudoelasticity of NiTi alloy could effectively improve the sealing performance of the NiTi alloy gasket. At the same time, the corrugated structure solved the problem that NiTi alloy flat gaskets need too much preload to form the initial seal. The relationship between contact stress and compression amount during the compression recovery process of gaskets is a key index to evaluate the mechanical properties of gaskets. It is also the key parameter to establishing the calculation model of gasket leakage rate and the design method of bolt flange joint. Therefore, this paper designed a NiTi alloy corrugated gasket, as shown in Figure 6. The influence of gasket structure parameters on the gasket’s mechanical properties was discussed based on the proposed NiTi corrugated gasket.

The finite element model is shown in Figure 7a. In this model, the upper and lower loading heads were simplified as rigid bodies. The lower loading head was fixed. The upper loading head was controlled by displacement. Its boundary condition was UX = UY = URX = URY = URZ = 0, and UZ was set to 0.1H and 0.15H, respectively. The penalty function controlled the contact property between the corrugated gasket and loading heads, and the friction coefficient was 0.15. The element type of the gasket was C3D8R. In order to ensure the effectiveness and accuracy of the model, at least two layers of the element were required along the gasket material thickness direction, as shown in Figure 7b. For subsequent analysis, the contact wave peaks of the corrugated gasket and upper flange were numbered as 1, 2, and 3, respectively. The trough numbers of the corrugated gasket in contact with the lower flange were 4, 5, 6, and 7, respectively.

## 3. Results

### 3.1. Verification of the Finite Element Analysis Method of NiTi Alloy Corrugation

At room temperature, NiTi alloy material is in a martensite state. The compression-resilience load–displacement curves of the corrugated sample are shown in Figure 8a. The temperature recovery curve of the corrugated specimen is shown in Figure 8b. The ultimate load of the NiTi alloy corrugated specimen is 229.08 N, the residual deformation of the specimen is 0.85 mm, and the resilience rate is 12.2%. The simulated ultimate load is 213.4 N, and the residual deformation is 0.87 mm. The resilience rate is 12.15%. The relative errors are 5.78%, 2.35%, and 0.41%, compared with the test results, respectively. All the errors are less than 10%. These errors came from the deviations from the geometric model, material properties, and the constitutive model. As can be seen from Figure 8b, the load increased gradually with the temperature increasing. The maximum load measured in the experiment at the heating stage was 145.17 N, and the maximum load calculated by the finite element method was 154.06 N. The relative error was 5.77%. The error might come from the material properties, the rounding error in the solution of the constitutive model, and the friction coefficient between the sample and the fixture surface.

It can be seen from Figure 8a that the compression resilience curves of NiTi alloy corrugated specimens at room temperature can be divided into four stages. The deformation process and phase transition characteristics are shown in Figure 9. The first stage (section AB in Figure 8a) is the linear elastic deformation stage of the corrugated specimens. As the displacement increases, the load exhibits a linear increase. When the displacement reaches 0.5 mm, the curves enter the second stage. At this time, the wave crest position of the corrugated specimens deformed. The arc radius of the crest gradually increased. The wave crest became the place where the stress was the greatest, and the martensite detwinning process occurred. The stress-induced transition of multiple martensite variants to a single martensite variant. The second stage (section BC) is the large deformation stage of NiTi alloy corrugation. With the displacement increasing, the load did not change a lot. However, the specimen deformed obviously along the thickness direction. The deformation modes were mainly the opening and diameter changing at the peak and trough and the bending deformation at the straight edge. The results showed that the stress increased slightly at the peaks and troughs, while the stress increased obviously at the straight edge. During the deformation process, the peaks and troughs of the corrugation undergo the further detwinning process. The content of stress-induced martensitic transition increases gradually. The detwinning also occurred on the straight edge. The third stage (section CD) was the small deformation stage of corrugated specimens in the unloading process. With the beginning of the unloading process, the deformation of the corrugated specimen along the thickness direction decreased slightly, while the load decreased sharply. The results showed that the peaks and troughs of the corrugated specimen have local recovery deformation. The radius of the contact area between the peak and the loading head decreased gradually. The results showed that the stress at the peak area decreased gradually. The reverse process of detwinning occurred in some martensites. In other words, single martensite transformed into a variety of martensitic variants, and the content of stress-induced martensitic decreased. The fourth stage (section DE) was the large deformation stage of the corrugated specimen during unloading. With the continuation of the unloading process, the opening displacement of the NiTi corrugation decreased gradually, and the load decreased linearly. The stress of the corrugation decreased gradually. A certain amount of residual stress existed at the wave peak, resulting in the incomplete reverse phase transition of martensite detwinning. A certain amount of stress-induced martensite variation existed at the wave peak, resulting in residual deformation of NiTi alloy corrugated.

As can be seen from Figure 8b, the load–temperature curves could be divided into two stages. The first was the section EF in Figure 8b. It increased approximately linearly with the temperature increasing. With the temperature increasing, the austenitic transition occurred in NiTi alloy. Martensite transformed towards austenite, and the martensite content gradually decreased. During the phase transition, the angle of the corrugated specimen gradually decreased and returned to its original shape. Because the relative position of the upper and lower loading head was fixed during the heating process, the load increased gradually. The second stage was section FG in Figure 8b. As the temperature increased to 95 °C, the load no longer increased gradually. This is due to the transition of martensite to austenite with the temperature increasing. As the upper and lower loading heads were fixed, the stress at the wave peak of the corrugated specimen increased, and the stress-induced martensitic transition took place there. When the temperature exceeded 95 °C, the NiTi alloys’ shape memory effect and pseudoelasticity reached equilibrium, resulting in a constant load.

In conclusion, according to the load–displacement curves and the load–temperature curves of NiTi alloy corrugated specimens, the experimental and finite element calculation results show a good agreement. The relative errors are less than 10%. The phase transition characteristics of NiTi alloy can better reveal the deformation mechanism of NiTi alloy corrugated samples, which indicates that the finite element simulation has high accuracy.

### 3.2. Effect of Gasket Structure on Mechanical Properties

#### 3.2.1. Plate Thickness

When the plate thickness *t* = 0.3 mm, the gasket height *H* = 2 mm, and the corrugation pitch *p* = 3.7 mm, the contact stress distribution of each ripple of the corrugated gasket with loading displacement of 0.1 H and 0.15 H is shown in Figure 10. Among the upper peaks, the contact stress of No. 2 corrugation was the largest, which was 1.5 times the peak stress of No. 1 and No. 3. Among the No. 4–No. 7 wave troughs in contact with the lower flange, the contact stress of No. 5 wave trough was larger, followed by No. 6. Because the corrugated gasket’s inner and outer reinforcing rings had fewer constraints on the radial deformation, and the No. 4 and No. 7 troughs slid along the radial direction, the No. 4 and No. 7 troughs failed to contact the lower flange fully. At the same time, the No. 1 and No. 3 wave peaks deformed along the thickness direction of the gasket driven by the radial deformation of No. 4 and No. 7 wave troughs, resulting in minor contact stress. However, No. 2, No. 5, and No. 6 were greatly constrained by adjacent corrugations, and their small radial deformation caused large contact stress.

It could be seen from the contact stress distribution of each crest and trough that crest No. 2 crest and trough No. 5 were the main sealing parts of the corrugated gasket. They were the key to the sealing design of the corrugated gasket. As can be seen from Figure 10, the contact stress of each crest and trough of the corrugated gasket after heating up was twice that after loading. This is because, during the loading process of the NiTi alloy corrugated gasket, the gasket’s state was below the austenitic transition temperature. Martensitic detwinning deformation occurred on the material at the crests and troughs under the compression load. When the gasket was heated, austenite transition occurred in NiTi alloy, that is, the shape memory effect.

Under the phase change driving force, the corrugated gasket contacted the flange more fully, and the contact stress increased greatly. This indicated that NiTi alloy corrugated pad could effectively reduce bolt load when it is pre-tightened below the initial temperature of austenitic transition and served above the end temperature of austenitic transition. At the same time, the reliability of gasket sealing was improved.

When the gasket height *H* = 2 mm and the corrugation pitch *p* = 3 mm, the contact stress variation in NiTi alloy corrugated gaskets with different plate thicknesses is shown in Figure 11. With the increase in NiTi alloy plate thickness, the contact stress of crest No. 2 and trough No. 5 gradually increased. With the increase in plate thickness, the contact stress of the NiTi alloy gasket after loading increased slightly, and the contact stress after heating increased greatly. The analysis showed that martensitic detwinning deformation occurred in the NiTi alloy corrugated gasket during the loading process. When the plate thickness was 0.3 mm and 0.4 mm, the deformations of the corrugated gasket were mainly bending of the straight edge and the crests opening during the loading process. When the plate thickness increased to 0.5 mm and 0.6 mm, the bending stiffness of the straight edge was greatly improved. The bending deformation of the straight edge was restrained, and the main deformation modes of the gasket are opening deformation and local deformation at the wave crest, as shown in Figure 12. The change in deformation mode made the contact stress of the corrugated gasket change. In addition, with the increase in plate thickness, the mass of NiTi alloy gasket increased, which would enhance the shape memory effect of NiTi alloy, leading to the increase in contact stress of NiTi alloy corrugated gasket after heating.

#### 3.2.2. Gasket Height

When the plate thickness *t* = 0.5 mm, the corrugation pitch *p* = 3 mm, and the compression displacement is 0.2 mm, the contact stress variations in the NiTi alloy corrugated gaskets with different gasket heights are shown in Figure 13. As shown in Figure 13, the contact stress at the contact area of the crest and trough increased first and then decreased with the gasket height *H* increasing. For the contact area of the crest, the contact stress was the largest when the gasket height *H* = 3 mm under compression. Under the heating condition, the contact stress was the largest when the gasket height *H* = 4 mm. For the contact area of the trough, the contact stress was the largest when the gasket height *H* = 3 mm under both compression and heating conditions. This was because the deformation mechanism of NiTi alloy corrugated gasket changed with the gasket height increasing. When the gasket height *H* = 2 mm, the main deformation modes of NiTi alloy corrugated gasket were opening deformation and local deformation at the wave crest. When the gasket height *H* = 3 mm, the main deformation mode changed to local deformation at the crest. When the gasket height is *H* ≥ 4 mm, the main deformation mode of NiTi alloy corrugated gasket was the bending deformation at the straight edge, as shown in Figure 14. The transformation of this deformation mode led to the contact stress of NiTi alloy corrugated gasket increasing first and then decreasing with the gasket height increasing.

#### 3.2.3. Corrugation Pitch

When the plate thickness *t* = 0.5 mm, the gasket height *H* = 3 mm, and the compression displacement was 0.2 mm, the contact stress variations in NiTi alloy corrugated gaskets with different corrugation pitches are shown in Figure 15. The contact stress of NiTi alloy corrugated gasket increased first and then decreased with the increase in corrugation pitch. When the corrugation pitch was 3 mm, the contact stress of the NiTi alloy corrugated gasket reached the maximum. The relationship between gasket contact stress and corrugation pitch was closely related to the corrugated gasket’s deformation. The deformation process of NiTi alloy gasket with different corrugation pitches is shown in Figure 16. When the corrugation pitch was less than 3 mm, the radial component force was small due to the small angle of the ripple, which is not enough to overcome the friction force. As a result, the main deformation mode of NiTi alloy corrugated gasket was local deformation of the crest at the contact area, and the contact stress was greater than that of the corrugated gaskets with the corrugation pitch greater than 3 mm. When the corrugation pitch was small, the deformation of the middle corrugation was greatly affected by the bending deformation of the corrugations on both sides. The increase in the corrugation pitch would restrain this interference phenomenon. Therefore, when the corrugation pitch was less than 3 mm, the contact stress increased gradually with the increase in the corrugation pitch. When the corrugation pitch was greater than 3 mm, the corrugation’s radial component force increased, making it difficult for the friction force to restrain the opening deformation of the corrugation. The main deformation mode of NiTi alloy corrugated gasket changed from the local deformation at the contact area to the opening deformation at the crest. With the increase in corrugation pitch, the opening displacement of the corrugation increased, resulting in a gradual decrease in contact stress. When the corrugation pitch was 3 mm, the component forces of the corrugated gasket in the radial direction and axial direction were equal. The radial component force was not enough to overcome the friction force. The axial component force could inhibit the local deformation of the gasket and make the NiTi alloy corrugated gasket bear the compressive load as a whole. Therefore, when the corrugation pitch was 3 mm, the contact stress of the NiTi alloy corrugated gasket was the largest.

## 4. Discussion

There are many deformation models in the service process of NiTi alloy corrugated gasket, such as crest opening deformation, bending deformation of the straight edge, local deformation at the contact area, and the combination of various deformation modes. These deformation modes are closely related to the geometry of gasket structure, which makes the gasket contact stress distribution show a complex change law. At the same time, structural deformation and detwinning phase transition of NiTi alloy mainly occurred in the preloading stage during the service process. Based on the principle of equivalent mechanics of continuous medium, the contact stress calculation model of NiTi alloy corrugated gasket was established, as shown in Equation (1).
(1)$$S_{G\_PRE}=E_{equal}\times \varepsilon_{equal}$$
where *S_G_PRE_* (MPa) is the contact stress of NiTi alloy corrugated gasket. *E_equal_* (MPa) is the equivalent elastic modulus of NiTi alloy corrugated gasket. *ε_equal_* is the equivalent strain of NiTi alloy corrugated gasket under preloading conditions.

The equivalent elastic modulus *E_equal_* of NiTi alloy corrugated gasket is affected by the gasket’s structural geometries and material mechanical properties. It is related to the plate thickness *T*, gasket height *H*, corrugation pitch *P*, and the elastic modulus *E_m_* of the martensitic phase of NiTi alloy corrugated gasket. According to the π theorem, *E_equal_* can be calculated by the following equation:(2)$$E_{equal}=f\left(\frac{T}{H},\frac{P}{H}\right)\times E_{m}$$
where $f\left(\frac{T}{H},\frac{P}{H}\right)$ is the correction coefficient of equivalent elastic modulus of NiTi alloy corrugated gasket, and its expression is shown in Equation (3).
(3)$$f\left(\frac{T}{H},\frac{P}{H}\right)=A_{0}+A_{10}\cdot \frac{T}{H}+A_{01}\cdot \frac{P}{H}+A_{20}\cdot {\left(\frac{T}{H}\right)}^{2}+A_{11}\cdot \frac{T}{H}\cdot \frac{P}{H}+A_{02}\cdot {\left(\frac{P}{H}\right)}^{2}$$
where A_0_, A_10_, A_20_, A_01_, A_02,_ and A_11_ are constants.

The equivalent strain of NiTi alloy corrugated gasket *ε_equal_* can be calculated by the following formula:(4)$$\varepsilon_{equal}=\frac{D_{G}}{H}$$
where *D_G_* (mm) refers to the axial deformation of the gasket during preloading.

Under the operating condition, the austenitic transition occurred on the NiTi alloy corrugated gasket. The stress-induced martensitic transition would also occur in the contact area locally. The phase transition is the driving force for the further deformation of NiTi alloy corrugated gasket under operating conditions. The upper and lower loading heads constrained the gasket and the corrugated gasket deformed under compressive load. The phase transition brings an extra deformation into the complex system. The contact stress also changed because of the phase transition. Under the operating condition, the contact stress of NiTi alloy corrugated gasket is related to material phase transition and contact stress under pre-tightening. The relationship could be expressed as follow:(5)$$S_{G\_OPT}=f\left(\frac{E_{a}}{E_{m}},\frac{T}{H},\frac{P}{H}\right)\times S_{G\_PRE}$$
where $f\left(\frac{E_{a}}{E_{m}},\frac{T}{H},\frac{P}{H}\right)$ is the correction coefficient considering NiTi phase transition behavior, as shown in Equation (6). *E_a_* (MPa)is the elastic modulus of the austenitic phase of NiTi alloy.
(6)$$f\left(\frac{E_{a}}{E_{m}},\frac{T}{H},\frac{P}{H}\right)={\left(\frac{E_{a}}{E_{m}}\right)}^{B_{0}+B_{10}\cdot \frac{T}{H}+B_{01}\cdot \frac{P}{H}+B_{20}\cdot {\left(\frac{T}{H}\right)}^{2}+B_{11}\cdot \frac{T}{H}\cdot \frac{P}{H}+B_{02}\cdot {\left(\frac{P}{H}\right)}^{2}}$$
where B_0_, B_10_, B_20_, B_01_, B_02_, and B_11_ are constants.

Using the equations above and fitting the data listed in Table 3, the contact stress of NiTi alloy corrugated gasket under preloading and operating conditions were obtained, as shown in Equation (7) and Equation (8), respectively.
(7)$$\begin{array}{l} S_{G\_PRE}=0.01133-1.745\cdot \frac{T}{H}+0.5562\cdot \frac{P}{H}-0.5857\cdot {\left(\frac{T}{H}\right)}^{2}\\ \;\;+1.558\cdot \frac{T}{H}\cdot \frac{P}{H}-0.4022\cdot {\left(\frac{P}{H}\right)}^{2}\times E_{m}\times \frac{D_{G}}{H} \end{array}$$
(8)$$S_{G\_OPT}={\left(\frac{E_{a}}{E_{m}}\right)}^{7.727-5.537\cdot \frac{T}{H}-7.188\cdot \frac{P}{H}-49.79\cdot {\left(\frac{T}{H}\right)}^{2}+26.11\cdot \frac{T}{H}\cdot \frac{P}{H}+0.444\cdot {\left(\frac{P}{H}\right)}^{2}}\times S_{G\_PRE}$$

As shown in Table 3, compared with the finite element calculation results, the maximum error of the contact stress calculation model of NiTi alloy corrugated gasket under preloading conditions was 16.1%. The maximum error of the contact stress under operating conditions was 11.2%. By analysing the causes, the sources of errors mainly include two aspects. The errors introduced by the material constitutive relationship and mesh in the simulation process. Second, the error introduced by the numerical fitting method and the rounding error caused by computer calculation. However, the maximum error of the calculation model was less than 20%, which can meet the needs of engineering applications.

## 5. Conclusions

NiTi alloy corrugated gasket has excellent mechanical properties of NiTi alloy, good compression resilience of corrugated structure, and high line seal specific pressure. It is widely used in the chemical industry, nuclear energy, and other fields. In this paper, the mechanical properties of NiTi alloy corrugated gasket were studied, and the conclusions are as follows:

(1) The finite element simulation method of NiTi alloy corrugated gasket considering shape memory effect, hyperelastic effect, and plastic deformation was established. It revealed the phase transition mechanism and deformation mechanism of NiTi alloy corrugated gasket during preloading and operation. The errors of key mechanical properties were less than 10%;

(2) During the service of NiTi alloy corrugated gasket, the shape memory effect can greatly improve the gasket contact stress. Additionally, the contact stress of NiTi alloy corrugated gasket under operation was twice that under preloading. The corrugated structure could effectively reduce the preload and improve the sealing performance;

(3) Plate thickness, gasket height, and corrugation pitch significantly influence the contact stress of NiTi alloy corrugated gasket, and show high nonlinear characteristics;

(4) Based on the continuous medium equivalence principle and π theorem, the NiTi alloy corrugated gasket’s contact stress calculation model was established. By comparing with the finite element results, the maximum error of the calculation results under the preloading condition is 16.1%. The maximum error of the calculation results under the operating condition is 11.2%.

## Figures and Tables

**Figure 1 materials-15-04836-f001:**
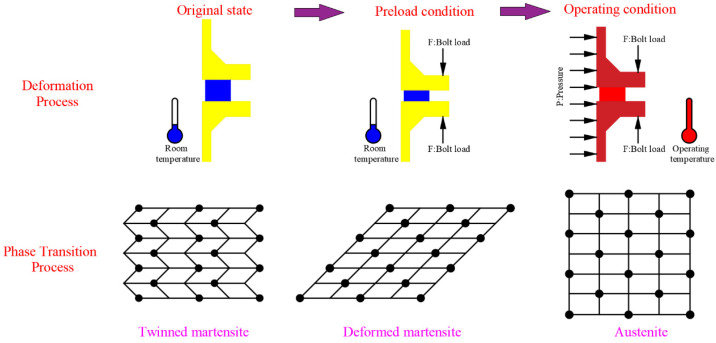
Deformation and phase transition process of NiTi alloy gasket.

**Figure 2 materials-15-04836-f002:**
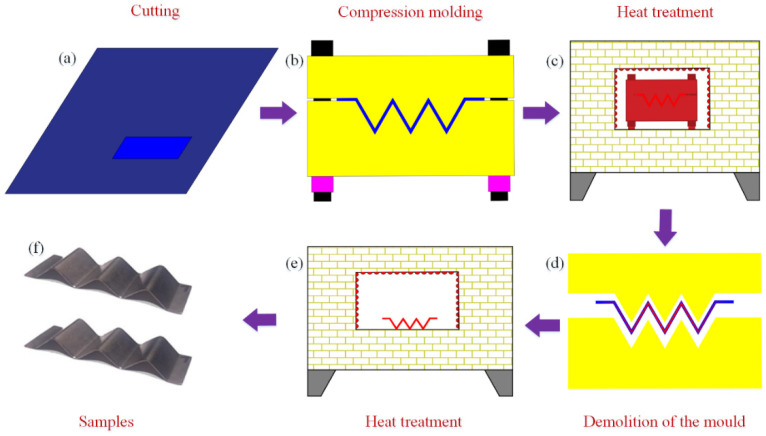
Corrugated samples preparation process: (**a**) take sample, (**b**) compression molding, (**c**) heat treatment, (**d**) demoulding, (**e**) secondary heat treatment, (**f**) samples.

**Figure 3 materials-15-04836-f003:**
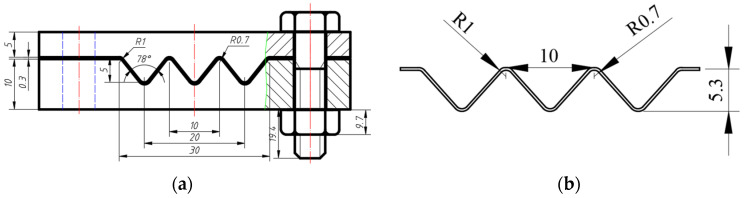
Geometries of corrugated sample and its processing fixture: (**a**) Geometry of the mould; (**b**) Geometry of corrugated specimens.

**Figure 4 materials-15-04836-f004:**
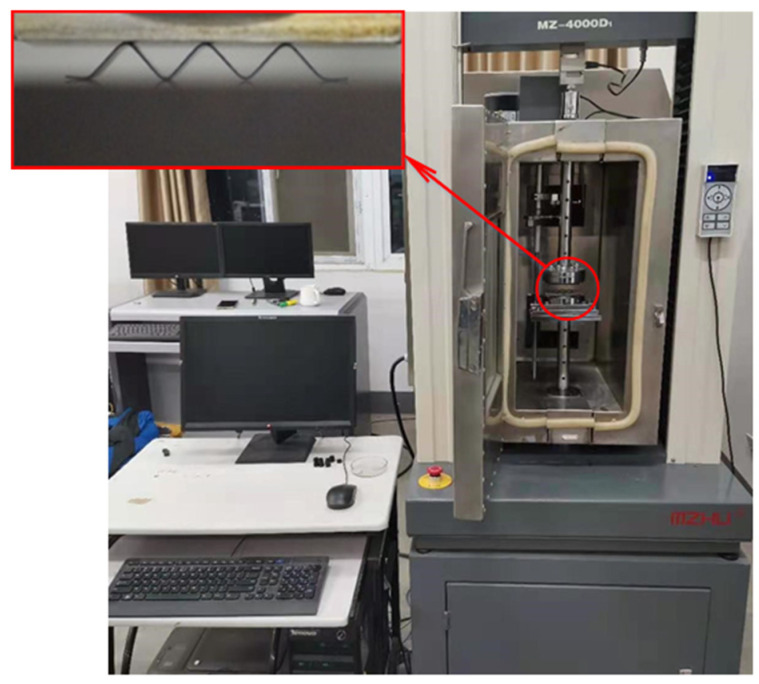
Mechanical property testing machine.

**Figure 5 materials-15-04836-f005:**
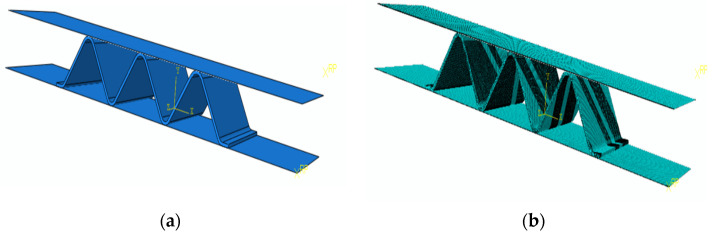
Finite element model of the NiTi alloy corrugation: (**a**) Geometrical model; (**b**) Geometrical model.

**Figure 6 materials-15-04836-f006:**
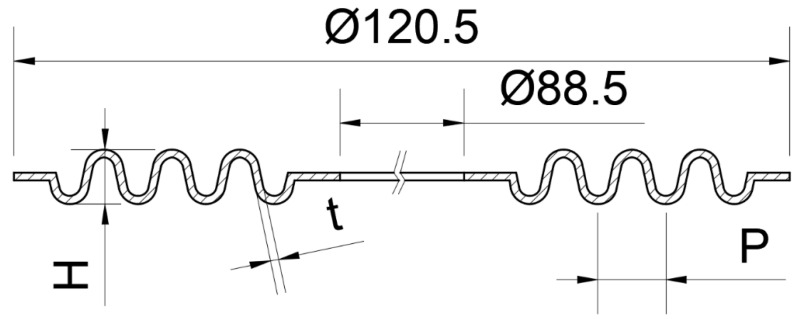
Structural diagram of NiTi alloy corrugated gasket: H, gasket height; t, plate thickness; P, corrugation pitch.

**Figure 7 materials-15-04836-f007:**
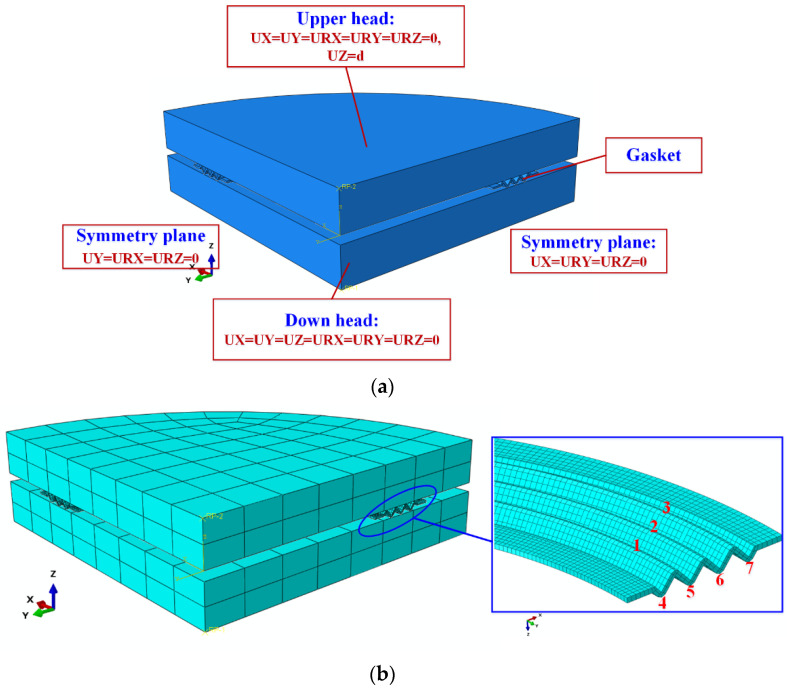
Finite element analysis model of mechanical properties of NiTi alloy corrugated gasket: (**a**) Boundary conditions; (**b**) Mesh model.

**Figure 8 materials-15-04836-f008:**
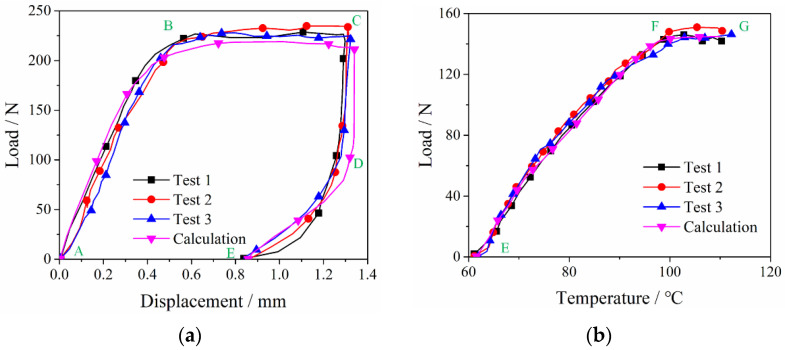
Experimental and simulation results of NiTi alloy corrugated sample: (**a**) Load–displacement curves at room temperature; (**b**) Load–temperature curve at elevated temperature.

**Figure 9 materials-15-04836-f009:**
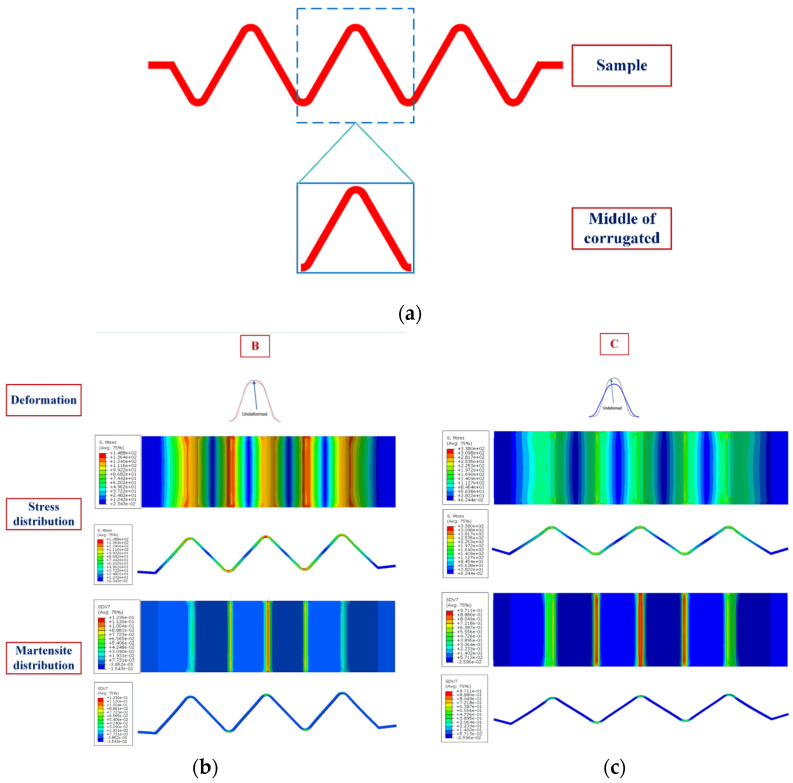
Deformation and phase transition characteristics of NiTi alloy corrugated sample: (**a**) Sample section; (**b**) Deformation process of point B in Figure 8; (**c**) Deformation process of point C in Figure 8; (**d**) Deformation process of point D in Figure 8; (**e**) Deformation process of point E in Figure 8; (**f**) Deformation process of point F in Figure 8; (**g**) Deformation process of point G in Figure 8.

**Figure 10 materials-15-04836-f010:**
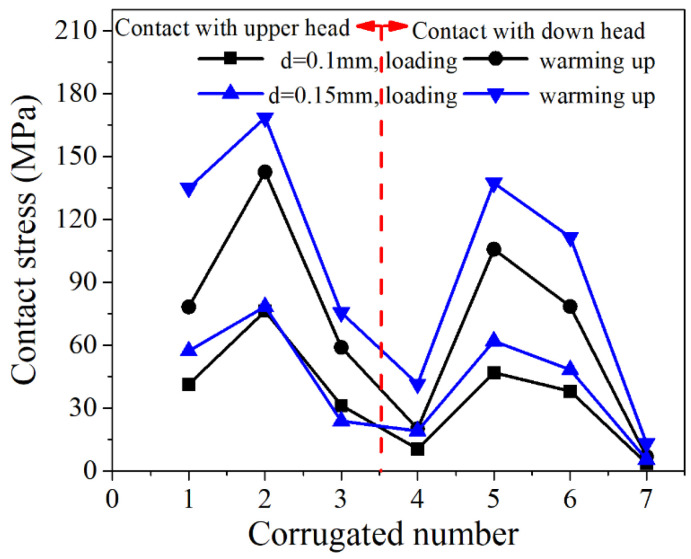
Stress in the contact area of corrugated gasket under different loading displacement when the plate thickness is 0.3 mm.

**Figure 11 materials-15-04836-f011:**
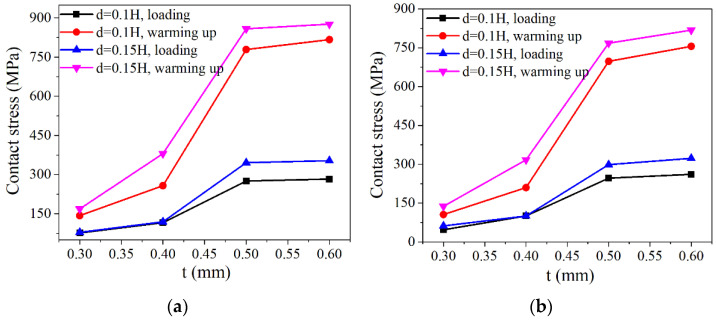
Contact stress of NiTi alloy corrugated gasket with different plate thickness: (**a**) Crest No. 2; (**b**) Trough No. 5.

**Figure 12 materials-15-04836-f012:**
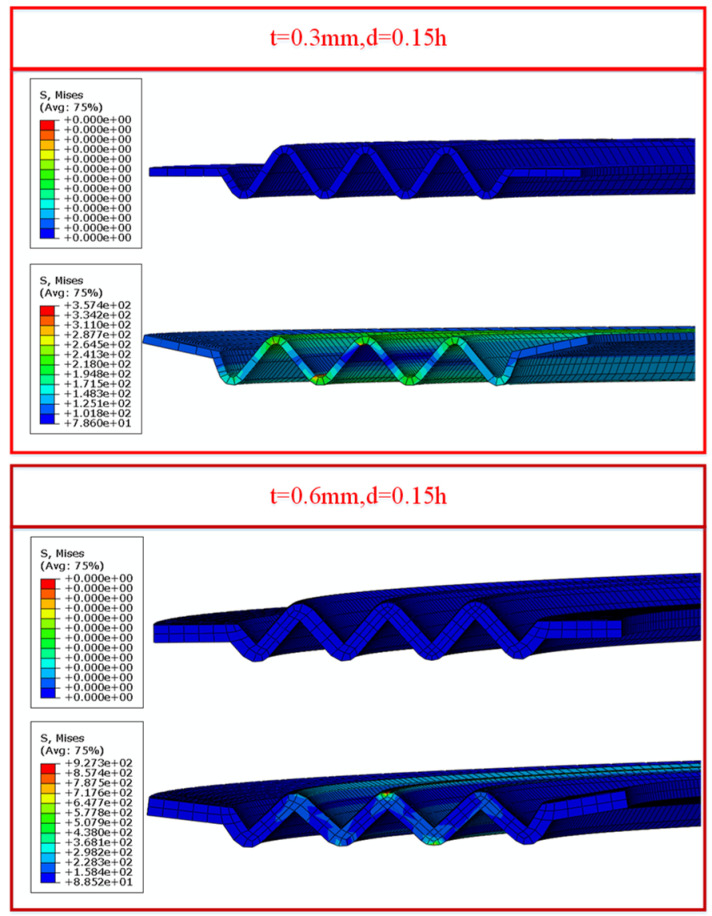
Deformation characteristics of NiTi alloy corrugated gasket with different plate thickness.

**Figure 13 materials-15-04836-f013:**
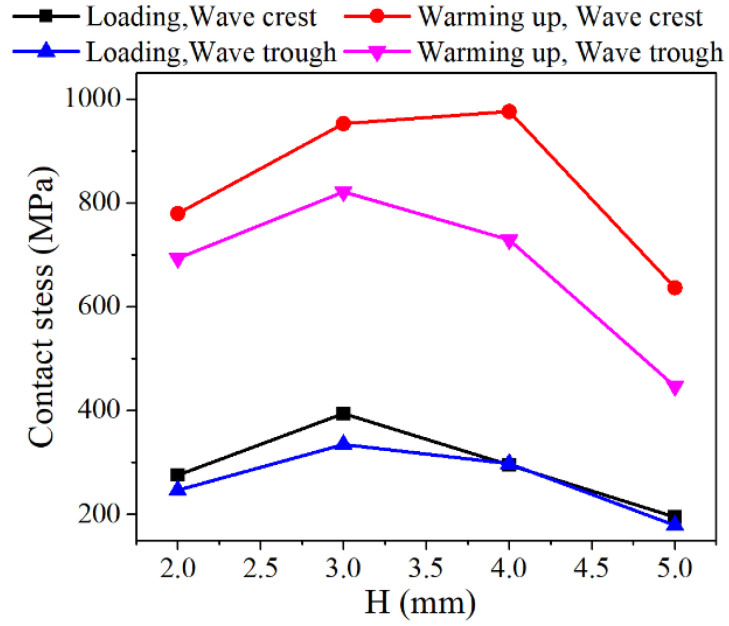
Relationship between contact stress and gasket height of NiTi alloy corrugated gasket.

**Figure 14 materials-15-04836-f014:**
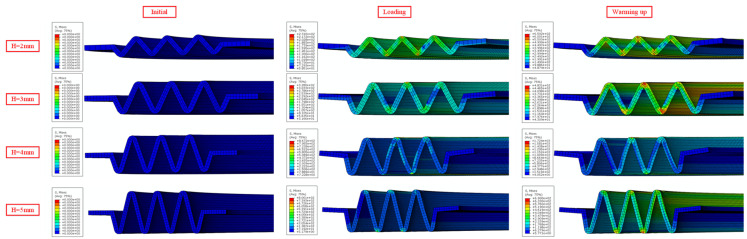
Deformation contours of NiTi alloy corrugated gasket with different heights.

**Figure 15 materials-15-04836-f015:**
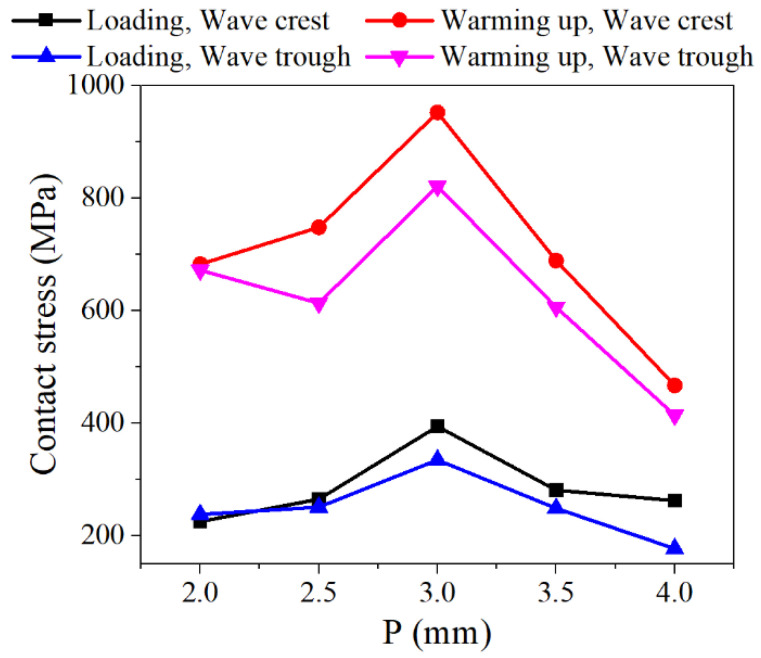
Contact stress distribution of NiTi alloy gasket with different corrugation pitches.

**Figure 16 materials-15-04836-f016:**
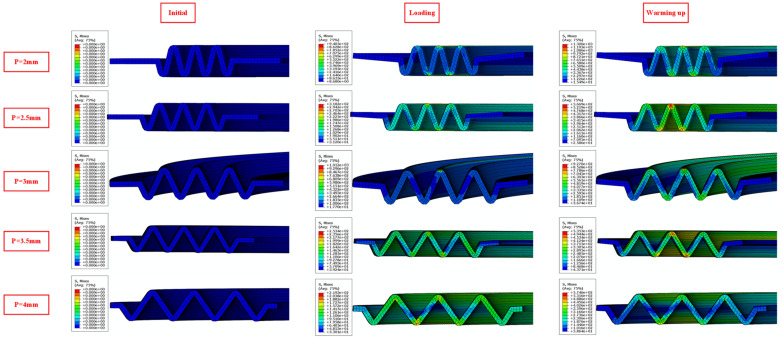
Deformation contours of NiTi alloy corrugated gasket with different corrugation pitches.

**Table 1 materials-15-04836-t001:** Chemical composition of NiTi alloy plate (wt.%) [21,30].

Ni	Co	Cu	Cr	Fe	Nb	C	H	O	N	Ti
55.59	0.005	0.005	0.005	0.012	0.005	0.046	0.001	0.03	0.001	margin

**Table 2 materials-15-04836-t002:** Physical parameters of NiTi alloy.

Physical Quantity	Value
Density/(g/cm^3^)	6.45
Young’s modulus of pure martensite/GPa	45
Young’s modulus of pure austenite/GPa	61
Poisson’s ratio	0.33
Influence coefficient of martensite/(MPa/K)	15.8
Influence coefficient of austenite/(MPa/K)	15.8
Maximum residual strain	0.023
Plastic limit/MPa (120 °C)	521
Plastic limit/MPa (20 °C)	618

**Table 3 materials-15-04836-t003:** Comparison between the results of finite element and calculation model with different structural parameters.

No.	*T* (mm)	*H* (mm)	*P* (mm)	*D_G_* (mm)	*S_G_PRE_* (MPa)			*S_G_OPT_* (MPa)		
					FEM	Theoretical Result	Relative Error (%)	FEM	Theoretical Result	Relative Error (%)
1	0.3	2	3	0.2	76.24	73.36	3.8	142.62	129.49	9.2
2	0.6	2	3	0.2	282.81	295.05	4.3	817.35	869.13	6.3
3	0.5	3	3	0.2	394.04	353.68	10.2	952.41	888.72	6.7
4	0.5	5	3	0.2	194.62	204.09	4.9	636.50	708.02	11.2
5	0.5	3	2	0.2	225.01	208.15	7.5	682.62	646.92	5.2
6	0.5	3	3.5	0.2	280.73	325.90	16.1	689.12	744.67	8.1
7	0.5	3	4	0.2	262.13	231.08	11.8	466.97	483.76	3.6

## Data Availability

It was followed according to MDPI Research Data Policies.

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
