# Peer review of "Calculation Model of Mechanical and Sealing Properties of NiTi Alloy Corrugated Gaskets under Shape Memory Effect and Hyperelastic Coupling: I Mechanical Properties"

_materials, 2022, doi:10.3390/ma15144836_

Round 1
Reviewer 1 Report
1. In the Introduction,
“Shape memory alloy is a sealing material with great potential. Its pseudo lasticity and shape memory effect can effectively improve the reliability of bolt flange connection sealing under fluctuating load conditions.” It is not clear that why the shape memory effect can improve the reliability and why SMA shows great potential in sealing. Please explain it more in details.
2. Please include more latest NiTi related literature in the Introduction.
3. Figure 9,
Please delete the Chinese characters in the figure.
The stress color legends are not readable. Please modify that.
Please also provide the front view of the simulation, because it is not easy to understand with the top view only.
Since the Figure 9 is the simulation of the experiments in Figure 8, the Figure 8 and 9 can be merged into a large figure for better understanding.
4. Figure 12, 14, 16,
The color legends are not readable.
5. The authors studied different parameters of the NiTi gasket and analyzed the influences, but how can the NiTi gaskets compare with the regular gaskets?
6. For my understanding, the NiTi gaskets should have ranges for the bolt preloads and the working temperatures. Could you please some data or estimations on the ranges?
7. The authors studied the mechanical responses when the NiTi gasket was heated. However, in practice, the working temperature may fluctuate. Will the performances change as well, if the gaskets are at periodic heating and cooling?
Author Response
- In the Introduction, "Shape memory alloy is a sealing material with great potential. Its pseudo lasticity and shape memory effect can effectively improve the reliability of bolt flange connection sealing under fluctuating load conditions." It is not clear why the shape memory effect can improve the reliability and why SMA shows great potential in sealing. Please explain it more in detail.
The authors' Answer:
We sincerely appreciate the valuable comments. The shape memory effect of the shape memory alloy effectively guarantees the contact stress of the gasket sealing surface under operating conditions. The excellent super-elasticity of the shape memory alloy can avoid the plastic deformation of the gasket under the short-term overload and effectively ensure the spring back amount of the gasket. Therefore, shape memory alloy material is an ideal sealing material. The original is modified as follows:
"Shape memory alloy is a sealing material with great potential. Its pseudoelasticity can avoid the plastic deformation of the gasket under short-term overload action, and effectively ensure the spring back amount of the gasket. The shape memory effect can effectively guarantee the contact stress of the sealing surface of the gasket under operating conditions. So the shape memory alloy gasket can effectively improve the reliability of bolt flange connection sealing under fluctuating load conditions."
- Please include more latest NiTi related literature in the Introduction.
The authors' Answer:
We sincerely appreciate the valuable comments. New literature has been added to the revised manuscript.
- Akhil Bhardwaj, Mihir Ojha, Ashutosh Garudapalli, Amit Kumar Gupta. Microstructural, mechanical and strain hardening behaviour of NiTi alloy subjected to constrained groove pressing and ageing treatment, Journal of Materials Processing Tech. , 2021, 294, 117-132.
- Dongjie Jiang, Yao Xiao. Modelling on grain size dependent thermomechanical response of superelastic NiTi shape memory alloy, International Journal of Solids and Structures, 2021, 210-211, 170-182.
- Jéssica Dornelas Silva, Pedro Damas Resende, Paula Ribeiro Garcia, et al. Fatigue resistance of dual-phase NiTi wires at different maximum strain amplitudes, International Journal of Fatigue, 2019, 125, 97-100.
- Komarov, V., Khmelevskaya, I., Karelin, R. et al. Deformation Behavior, Structure and Properties of an Equiatomic Ti–Ni Shape Memory Alloy Compressed in a Wide Temperature Range, Trans Indian Inst Met, 2021,74, 2419–2426.
- Komarov V, Khmelevskaya I, Karelin R, et al. Deformation Behavior, Structure, and Properties of an Aging Ti-Ni Shape Memory Alloy after Compression Deformation in a Wide Temperature Range, JOM: the journal of the Minerals, Metals & Materials Society, 2021,5, 620-629.
- Figure 9, Please delete the Chinese characters in the figure. The stress color legends are not readable. Please modify that. Please also provide the front view of the simulation, because it is not easy to understand with the top view only. Since Figure 9 is the simulation of the experiments in Figure 8, Figure 8 and 9 can be merged into a large figure for better understanding.
The authors' Answer:
We sincerely appreciate the valuable comments. Figure 9 has been modified in the manuscript.
- Figure 12, 14, 16, The color legends are not readable.
The authors' Answer:
We sincerely appreciate the valuable comments. Figures 12, 14, and 16 have been modified in the manuscript.
- The authors studied different parameters of the NiTi gasket and analyzed the influences, but how can the NiTi gaskets compare with the regular gaskets?
The authors' Answer:
We sincerely appreciate the valuable comments. NiTi alloy flat gaskets are designed using the face seal principle. And NiTi alloy flat gasket is difficult to occur large deformation due to the high hardness, so the initial sealing conditions cannot be formed. This paper proposes NiTi alloy corrugated gaskets to address the problems of NiTi alloy flat gaskets. The design principle of multi-channel sealing is adopted to increase the contact stress between the NiTi alloy corrugated gasket and the flange sealing surface so that it can undergo plastic deformation, thereby forming the initial sealing condition. In addition, the sealing pressure ratio of multi-channel sealing is much larger than that of flat sealing in terms of the sealing principle. Therefore, the NiTi alloy corrugated gasket proposed in this paper can play the advantages of NiTi alloy more than the NiTi alloy flat gasket.
- For my understanding, the NiTi gaskets should have ranges for the bolt preloads and the working temperatures. Could you please some data or estimations on the ranges?
The authors' Answer:
We sincerely appreciate the valuable comments. The service temperature should be higher than the initial temperature of austenitic transformation for the NiTi alloy gasket. The best service temperature is higher than the end temperature of austenitic transformation. The service temperature range requires 77°C to 500°C for this article (the temperature at which solid solution occurs). The preload of the NiTi alloy corrugated gasket is directly related to the gasket's structural parameters and the medium's internal pressure. Generally, engineers calculate the bolt preload through the compression-resilience rate of the gasket. The amount of compression is about 30% for NiTi alloy gaskets. Therefore, the preload is the best when the amount of compression is 30% for the bolt flange structure using NiTi alloy corrugated gasket.
- The authors studied the mechanical responses when the NiTi gasket was heated. However, in practice, the working temperature may fluctuate. Will the performances change as well, if the gaskets are at periodic heating and cooling?
The authors' Answer:
We sincerely appreciate the valuable comments. This paper discusses the influence of the NiTi alloy shape memory effect on the sealing performance of corrugated gaskets. When the NiTi alloy is completely unloaded, the shape memory effect can ensure that the gasket has a certain sealing performance. As the temperature increases, the contact stress caused by the shape memory effect increases (Figure 8). However, when the temperature exceeds 100°C, the contact stress caused by the shape memory effect increases slowly. When the temperature fluctuates below 100°C, the shape memory effect has a greater impact on the sealing performance of the gasket. When the temperature fluctuates above 100°C, it has less impact on the sealing performance of the gasket. Therefore, to ensure the sealing performance, the minimum working temperature of NiTi alloy is the termination temperature of martensitic transformation, and the maximum temperature is the solution temperature of NiTi alloy material. The temperature fluctuates within this range, and the shape memory effect has little effect on sealing performance.

Reviewer 2 Report
The article « Calculation model of mechanical and sealing properties of NiTi alloy corrugated gaskets under shape memory effect and hyperelastic coupling: â… Mechanical properties» is devoted to a practical topic and focuses mainly on modeling, the experimental part, which could add a scientific sound, is unfortunately very small. However, the article may be published after revision.
The most significant conclusions made in the work should be added to the abstract.
Check the spelling of the alloy - NiTi (T - capital)
In the introduction, more attention should be paid directly to shape memory alloys based on titanium nickeline, the technology of their thermomechanical processing, and properties. You can refer to these articles: 10.1007/s12666-021-02355-x 10.1007/s11837-020-04508-7
It is necessary to indicate the reason for choosing the 500℃ heat treatment mode. This temperature was most likely chosen for stress relaxation (you can also use the reference above for this)
It is necessary to indicate why the mode was chosen as the heat treatment - 150 ℃ for 10 min
The chemical composition indicated in Table 1 corresponds to 50.53 at% Ni - as a rule, when working with these alloys, they speak of atomic percentages, not weight. This must be specified
The results of DSC raise questions - it follows from the text that the transformation took place in one phase - is it really so? At the same time, the hysteresis of the direct martensite transformation is quite wide. In an alloy containing more than 50.2 at % Ni, aging processes may occur.
Reverse martensitic transformation temperatures too high for 50.53 at% Ni alloy
I ask you to give the results of DSC before and after processing in the article.
As I already wrote in the article, a very weak experimental part. It will be good if you find an opportunity to add structural-phase studies and hardness measurements to this work.
The part related to modeling is written very well.
Author Response
- In the introduction, more attention should be paid directly to shape memory alloys based on titanium nickeline, the technology of their thermomechanical processing, and properties. You can refer to these articles: 10.1007/s12666-021-02355-x, 10.1007/s11837-020-04508-7
The authors' Answer:
We sincerely appreciate the valuable comments. We have reorganized the recent research work on shape memory alloys and added several representative literature, as follows:
- Akhil Bhardwaj, Mihir Ojha, Ashutosh Garudapalli, Amit Kumar Gupta. Microstructural, mechanical and strain hardening behaviour of NiTi alloy subjected to constrained groove pressing and ageing treatment, Journal of Materials Processing Tech., 2021, 294, 117-132.
- Dongjie Jiang, Yao Xiao. Modelling on grain size dependent thermomechanical response of superelastic NiTi shape memory alloy, International Journal of Solids and Structures, 2021, 210-211, 170-182.
- Jéssica Dornelas Silva, Pedro Damas Resende, Paula Ribeiro Garcia, et al. Fatigue resistance of dual-phase NiTi wires at different maximum strain amplitudes, International Journal of Fatigue, 2019, 125, 97-100.
- Komarov, V., Khmelevskaya, I., Karelin, R.et al. Deformation Behavior, Structure and Properties of an Equiatomic Ti–Ni Shape Memory Alloy Compressed in a Wide Temperature Range, Trans Indian Inst Met,2021,74, 2419–2426.
- Komarov V , Khmelevskaya I , Karelin R , et al. Deformation Behavior, Structure, and Properties of an Aging Ti-Ni Shape Memory Alloy after Compression Deformation in a Wide Temperature Range, JOM: the journal of the Minerals, Metals & Materials Society, 2021,5, 620-629.
- It is necessary to indicate the reason for choosing the 500℃ heat treatment mode. This temperature was most likely chosen for stress relaxation (you can also use the reference above for this)
The authors' Answer:
We sincerely appreciate the valuable comments. In the part of corrugated gasket processing, the author tested the effect of different setting temperatures on the dimensional accuracy and compression-resilience characteristics of NiTi alloy corrugated gasket samples and found that 500°C was the best temperature for setting. See the literature for details:
[21] Zhan Y, He L, Lu X, et al. The Effect of Ageing Treatment on Shape-Setting and Shape Memory Effect of a NiTi SMA Corrugated Structure[J]. Advances in Materials Science and Engineering. 2020, 2020(17): 1-11.
- It is necessary to indicate why the mode was chosen as the heat treatment - 150 ℃for 10 min
The authors' Answer:
We sincerely appreciate the valuable comments. The processing parameters are from the previous work of the research group. See reference [21] for details.
[21] Zhan Y, He L, Lu X, et al. The Effect of Ageing Treatment on Shape-Setting and Shape Memory Effect of a NiTi SMA Corrugated Structure[J]. Advances in Materials Science and Engineering. 2020, 2020(17): 1-11.
- The chemical composition indicated in Table 1 corresponds to 50.53 at% Ni - as a rule, when working with these alloys, they speak of atomic percentages, not weight. This must be specified
The authors' Answer:
We sincerely appreciate the valuable comments. We tested the composition of the material using Optical Emission Spectrometer and determined to be the mass fraction. The same test results can be seen in references [21] and [26]:
[21] Zhan Y, He L, Lu X, et al. The Effect of Ageing Treatment on Shape-Setting and Shape Memory Effect of a NiTi SMA Corrugated Structure[J]. Advances in Materials Science and Engineering. 2020, 2020(17): 1-11.
[26] Liang He, Xiaofeng Lu, Xiaolei Zhu, Qing Chen. T Influence of Structural Parameters of Shape Memory Alloy Corrugated Gaskets on the Contact Pressure of Bolted Flange Joints [J]. Advances in Materials Science and Engineering, 2021
- The results of DSC raise questions - it follows from the text that the transformation took place in one phase - is it really so? At the same time, the hysteresis of the direct martensite transformation is quite wide. In an alloy containing more than 50.2 at % Ni, aging processes may occur.
The authors' Answer:
We sincerely appreciate the valuable comments. The martensitic transformation process is very complex. In addition, martensitic variants are varied and numerous. Many scholars have done a lot of research work in this area. The existence of martensitic variants is the main reason for the hysteresis of phase transformation. However, in this paper, its constitutive relation model only considers the B19 variant and adopts a two-step phase transformation in studying the NiTi alloy corrugated gasket.
- Reverse martensitic transformation temperatures too high for 50.53 at% Ni alloy I ask you to give the results of DSC before and after processing in the article.
The authors' Answer:
We sincerely appreciate the valuable comments. The main work of this paper is to discuss the deformation mechanism of the NiTi alloy corrugated gasket. The material processing method in this paper is derived from the research work of other students in the research group. For details, please refer to the literature:
[21] Zhan Y, He L, Lu X, et al. The Effect of Ageing Treatment on Shape-Setting and Shape Memory Effect of a NiTi SMA Corrugated Structure[J]. Advances in Materials Science and Engineering. 2020, 2020(17): 1-11.
- As I already wrote in the article, a very weak experimental part. It will be good if you find an opportunity to add structural-phase studies and hardness measurements to this work.
The authors' Answer:
We sincerely appreciate the valuable comments. We will seriously consider this part of the work in the follow-up research process.
Reviewer 3 Report
It is correct scientific work of NiTi alloy.However, I consider that its publication in a journal with a high impact factor such as Materials is not relevant.
There are many articles about NiTi. Its properties and characteristics are well known.
It should be recognized that the novelty of the manuscript lies in the analysis and simulation of corrugated gaskets.
However, the simulated corrugation is highly idealized.
Regarding the concordance between the theoretical model and the simulation, this is very sensitive to the parameters used in both cases, and it is easily possible to obtain a good concordance.
The same comment regarding the experimental results.
Author Response
It is correct scientific work of NiTi alloy.
However, I consider that its publication in a journal with a high impact factor such as Materials is not relevant.
There are many articles about NiTi. Its properties and characteristics are well known.
It should be recognized that the novelty of the manuscript lies in the analysis and simulation of corrugated gaskets.
However, the simulated corrugation is highly idealized.
Regarding the concordance between the theoretical model and the simulation, this is very sensitive to the parameters used in both cases, and it is easily possible to obtain a good concordance.
The same comment regarding the experimental results.
The authors' Answer:
We are very grateful for your comments on the manuscript. This paper studies the mechanical properties of NiTi alloy corrugated gaskets by experiments and finite element methods. It is about applying NiTi alloy and meets the journal's scope.
Round 2
Reviewer 2 Report
In general, responses to comments and a new version of the article "Calculation model of mechanical and sealing properties of NiTi alloy corrugated gaskets under shape memory effect and hyperelastic coupling: â… Mechanical properties" can be published in the present form.
I hope the authors will continue their work, and the new article will be devoted to structural-phase and functional properties studies, which is also of interest. Good luck!
Author Response
We sincerely appreciate the reviewer‘s comments.
Reviewer 3 Report
The authors modify the manuscript taking into account the comments of the other two revierwers.